# GPR91 Receptor Mediates Protection against Doxorubicin-Induced Cardiotoxicity without Altering Its Anticancer Efficacy. An In Vitro Study on H9C2 Cardiomyoblasts and Breast Cancer-Derived MCF-7 Cells

**DOI:** 10.3390/cells9102177

**Published:** 2020-09-27

**Authors:** Matthieu Dallons, Esma Alpan, Corentin Schepkens, Vanessa Tagliatti, Jean-Marie Colet

**Affiliations:** Department of Human Biology & Toxicology, Faculty of Medicine and Pharmacy, University of Mons, Place du Parc 20, 7000 Mons, Belgium; matthieu.dallons@umons.ac.be (M.D.); esma.alpan@student.umons.ac.be (E.A.); corentin.schepkens@umons.ac.be (C.S.); vanessa.tagliatti@umons.ac.be (V.T.)

**Keywords:** doxorubicin, cardiotoxicity, H9C2, GPR91, succinate, *cis*-epoxysuccinate

## Abstract

Doxorubicin (DOX) is an anticancer drug widely used in oncology, especially for breast cancer. The main limitation of DOX treatment is its cardiotoxicity due to the cumulative dose. Clinically, DOX-induced cardiomyopathy develops as a progressive heart failure caused by a progressive cardiomyocyte’s death. For long, the oxidative stress induced by DOX was considered as the main toxic mechanism responsible for heart damage, but it is now controverted, and other processes are investigated to develop cardioprotective strategies. Previously, we studied DOX-induced cardiotoxicity and dexrazoxane (DEX), the only cardioprotective compound authorized by the FDA, by ^1^H-NMR metabonomics in H9C2 cells. We observed an increased succinate secretion in the extracellular fluid of DEX-exposed cardiomyocytes, a finding that led us to the hypothesis of a possible protective role of this agonist of the GPR91 receptor. The objective of the present work was to study the effect of succinate (SUC) and *cis-*epoxysuccinate (*cis*-ES), two agonists of the GPR91 receptor, on DOX-induced cardiotoxicity to H9C2 cells. To this purpose, several toxicity parameters, including cell viability, oxidative stress and apoptosis, as well as the GPR91 expression, were measured to assess the effects of DEX, SUC and *cis*-ES either alone or in combination with DOX in H9C2 cells. A ^1^H-NMR-based metabonomic study was carried out on cellular fluids collected after 24 h to highlight the metabolic changes induced by those protective compounds. Moreover, the effects of each agonist given either alone or in combination with DOX were evaluated on MCF-7 breast cancer cells. GPR91 expression was confirmed in H9C2 cells, while no expression was found in MCF-7 cells. Under such experimental conditions, both SUC and *cis*-ES decreased partially the cellular mortality, the oxidative stress and the apoptosis induced by DOX. The SUC protective effect was similar to the DEX effect, but the protective effect of *cis*-ES was higher on oxidative stress and apoptosis. In addition, the metabonomics findings pointed out several metabolic pathways involved in the cardioprotective effects of both GPR91 agonists: the stimulation of aerobic metabolism with glucose as the main fuel, redox balance and phospholipids synthesis. Finally, none of the GPR91 agonists jeopardized the pharmacological effects of DOX on MCF-7 breast cancer cells.

## 1. Introduction

Doxorubicin (DOX) is a widely used anticancer drug belonging to the anthracyclines family and used in the treatment of carcinomas, sarcomas and hematologic tumors. Breast cancer is one of the most carcinomas treated with DOX, especially when there is no indication of a targeted therapy. Nowadays, the main limitation of DOX treatment comes from its problematic cardiotoxicity [1,2]. The severity and irreversibility of the DOX-induced cardiotoxicity depends on the cumulated DOX doses and range from subclinical myopathy to severe heart failure, leading to the patient’s death in the worst cases [3]. Cardiotoxicity is believed to result from excessive oxidative stress induced by DOX in cardiomyocytes through the overproduction of reactive oxygen species (ROS) [1,4,5], which impair the mitochondrial function and membrane integrity [4]. Eventually, cardiomyocytes undergo increasing apoptosis and necrosis, clinically expressed by progressive heart failure [6].

These findings led to clinical trials evaluating the protective effects of some antioxidants to prevent DOX-induced cardiotoxicity. However, only dexrazoxane (DEX) demonstrated some significant protective effects in patients and was therefore approved by the FDA as a cardioprotective drug. DEX acts by reducing ROS production through iron chelation. DEX is intravenously injected into patients 10 to 30 min prior to the onset of DOX treatment at a recommended DEX/DOX dose ratio of 10:1 [4]. Clinically, the DEX cardioprotective effect has been studied for more than 20 years. There are evidences that the incidence of heart failure is significantly reduced in patients pretreated with DEX [7]. Despite its clear ability to reduce the incidence of heart complications in DOX-treated cancerous patients, DEX can be responsible for different adverse effects, particularly in pediatric patients. These findings led to a restriction of the indication of DEX for cardioprotection [4,8]. Therefore, the need for new cardioprotective strategies is obvious, hopefully safer and more efficacious than DEX.

Although the increased oxidative stress is considered as the main toxicological mechanism responsible of DOX cardiotoxicity, it remains unclear why, among all the antioxidants clinically tested, only DEX showed some substantial benefice. These unexpected findings led to the conclusion that the oxidative stress was most likely not the only toxic mechanism, and other causes should be investigated. Besides the beneficial effects of DEX on DOX-induced oxidative stress, some authors reported that DEX is also able to prevent its inhibition on the mitochondrial topoisomerase IIβ [9], another proposed cardiotoxicological mechanism [10]. This revived the search for other mechanisms of DOX-induced cardiotoxicity.

Besides their well-known roles in cell respiration [11], some Krebs cycle intermediates are suspected to play an active role in crucial cell activities. For instance, it was proposed that succinate could accumulate in the interstitial space in the case of ischemia [12]. In 2004, succinate was also pointed as the endogenous ligand of GPR91, a previously orphan G-protein-coupled receptor. Succinate was therefore considered to have unexpected signaling functions as an “alarming” signal able to trigger GPR91 [13]. Since then, the physiological functions and pathological implications of GPR91 have been studied. The receptor expression was found in many tissues and organs, including the heart [14]. However, its roles in cardiac physiology and pathophysiology are still unclear. While some evidences link it to cardiac hypertrophy through intracellular pathways involving cytoplasmic Ca^2+^, RAS proteins, mitogen-activated protein kinases (MAPK) and inositol 3,4,5-triphosphate (IP3) generated by phospholipase C [15], other findings associate GPR91 to the induction of apoptosis in cardiomyocytes through the activation of protein kinase A (PKA) [16]. Such opposite effects suggest that GPR91 stimulation could regulate the cardiomyocytes’ fate, promoting either cell death or cell growth and survival on purpose. Clearly, should GPR91 be considered as a pharmacological target in drug-induced cardiotoxicity, a better understanding of how GPR91 functions and how it is regulated is needed. The discovery of a GPR91-succinate interaction brings new perspectives for the development of new treatments to counteract different cardiomyopathies involving either a loss of cardiomyocytes or an excess of cardiomyocyte growth. We previously studied the effects of DOX and DEX on H9C2 cardiomyoblasts metabolism by ^1^H-NMR metabonomics. We highlighted that exposing cells to DEX prior to DOX caused an oversecretion of succinate in the culture medium. This led us to hypothesize that the release of succinate in DEX-exposed cells could protect against DOX-induced cardiotoxicity in H9C2 cells by an autocrine activation of the GPR91 receptor and the subsequent triggering of cell survival pathways [17].

In this study, we investigated the putative protective role of GPR91 against DOX-induced cardiotoxicity in rat H9C2 cells. In parallel, the effect of stimulating GPR91 on DOX anticancer efficacy was checked in breast cancer-derived MCF-7 cells.

## 2. Materials and Methods

### 2.1. Materials

The H9C2 (2-1) (ECACC 88092904) cell line was purchased from the European Collection of Authenticated Cell Cultures (Salisbury, United Kingdom). The MCF-7 cell line was obtained from ATCC (Manassas, VA, USA). Dulbecco’s modified Eagle’s medium (DMEM), trypsin-EDTA 0.05%, penicillin/streptomycin, L-glutamine, *cis-*epoxysuccinate (*cis-*ES) and Alexa Fluor^®^ 488 goat anti-rabbit immunoglobulin G (IgG) secondary antibody (AB) were acquired from Gibco (Thermo Fisher Scientific, Waltham, MA, USA). Dulbecco’s phosphate-buffered solution (D-PBS), phosphate-buffered solution (PBS), fetal bovine serum (FBS), doxorubicin hydrochloride (DOX), dexrazoxane (DEX), dichloro-dihydro-fluorescein diacetate (DCFH-DA), glutaraldehyde, triton, crystal violet, succinate and 4′,6-diamidino-2-phénylindole (DAPI) mounting solution were obtained from Sigma-Aldrich (Saint-Louis, MO, USA). Rabbit anti-GPR91 AB (BS2961) was purchased from BioWorld Technology (Bloomington, IN, USA).

### 2.2. Cell Culture

H9C2 cells were cultured in DMEM high glucose supplemented with L-Glutamine 2 mM, heat-inactivated FBS 10% and antibiotics (100 µg/mL of streptomycin and 100 unit/mL of penicillin), incubated at 5% CO_2_ and 37 °C, in a humidified atmosphere. Culture medium was replaced every 2 to 3 days, and cells were split when they reached 80% of confluence using trypsin-EDTA 0.05%. For viability and ROS quantification procedures, 30,000 cells were first seeded in 96-well plates and were kept growing during 24 h before any exposure. For the caspase 3 quantification assay, cells were seeded at a density of 30,000 cells/cm^2^ and were kept growing during 48 h before any exposure.

MCF-7 breast cancer cells were cultured in DMEM high glucose supplemented with L-glutamine 4 mM, heat-inactivated FBS 10% and antibiotics (100 µg/mL of streptomycin and 100 unit/mL of penicillin), incubated at 5% CO_2_ and 37 °C, in a humidified atmosphere. Culture medium was replaced every 2 to 3 days, and cells were split when they reached 100% of confluence using trypsin-EDTA 0.05%. For the proliferation assay, 30,000 cells were first seeded in 96-well plates and were kept growing during 24 h before any exposure.

### 2.3. Immunofluorescence Staining of GPR91

For immunostaining purposes, H9C2 and MCF-7 cells were grown on lamellae in 6-well plates. Cells were first washed with PBS and fixed with paraformaldehyde for 20 min. Cells were permeabilized using triton 0.05% for 15 min and then incubated during 20 min with 0.05% casein for blocking. After rinsing with PBS, cells were labeled with a specific rabbit primary anti-GPR91 AB diluted in a blocking solution (1:100) for 2 h at room temperature. Cells were washed with PBS and exposed to Alexa Fluor^®^ goat anti-rabbit IgG (secondary AB) during 1 h in the dark at room temperature. Cells were finally washed, and lamellae were mounted on slides with DAPI mounting solution and sealed with nail polish. A control without primary AB was made for both cell types. Coverslips were analyzed using a fluorescent microscope Olympic Fluoview FV1000 (Olympus, Berchem, Belgium). All images were obtained at a magnification of 60× by Z-projection.

### 2.4. Cell Viability Assay

The impacts of DOX, DEX, succinate (SUC) and *cis*-ES on the H9C2 cell viability were assessed by a crystal violet procedure in 96-well plates, as previously mentioned [17]. After cell exposure, the culture medium was removed, and cells were washed twice with PBS and incubated with 100 µL of glutaraldehyde 1% for 15 min at room temperature (RT). Glutaraldehyde was then removed, and 100 µL of crystal violet 1% was added for 30 min at RT. The plate was profusely washed with water and then dried at RT. The plate was incubated with 100 µL of triton 0.2% and agitated for 60 min. The absorbance was read at 570 nm with a VersaMax plate reader (Molecular Devices, Wokingham, England). Relative mean cellular viability was determined. All exposures were performed by diluting the compounds into a fresh culture medium. Cells were randomly assigned to different conditions in 96-well plates: a control condition receiving the vehicle; cells receiving DOX (0.3 µM); cells receiving either DEX, SUC or *cis*-ES (3 µM) and cells receiving either DEX, SUC or *cis-*ES 30 min prior to DOX exposure. Viability was measured after 24, 48 and 72 h of exposure. Moreover, increasing the doses of SUC or *cis-*ES (3–6–30–100 µM) and the combined exposures of either SUC or *cis*-ES with DEX were evaluated for their possible protective roles against DOX-induced cell mortality after 48 h of exposure.

### 2.5. Caspase 3 Activity Assay

The ability of DOX, DEX, SUC and *cis*-ES to induce or prevent apoptosis was indirectly assessed using the caspase 3 enzyme activity, as previously described [17]. H9C2 cells were randomly split in different conditions and were exposed to DOX 5 µM, DEX 50 µM, SUC 50 µM and *cis*-ES 50 µM in 96-well plates. Cells were also exposed once to each cardioprotective compound 30 min before DOX exposure, keeping a protective compound/DOX dose ratio of 10/1. After 4 h of exposure, the culture medium was removed. Cells were washed twice with PBS and then collected in PBS by scraping. A centrifugation at 260× *g* during 5 min was performed at 4 °C. The cell pellet was stored at −80 °C before further analysis. The activity of caspase 3 was evaluated using EnzChek Caspase-3 Assay Kit 1 (Thermo Fisher Scientific, Waltham, MA, USA), according to the manufacturer’s instructions. The intensity of fluorescence was read with a Glomax Explorer (Promega, Leiden, The Netherlands) plate reader at 342/441 nm (excitation/emission). Data were expressed as the relative mean of caspase 3 activity.

### 2.6. Oxidative Stress Quantification Assay

The effects of DOX, DEX, SUC and *cis-*ES on the redox state of H9C2 cells were indirectly assessed by a quantification of ROS production using a DCFH-DA probe [18]. Cells were randomly exposed in PBS to DOX 5 µM, DEX 50 µM, SUC 50 µM and *cis*-ES 50 µM in 96-well plates for 2 h. Cells were also exposed once to each cardioprotective compound 30 min before DOX exposure, keeping a protective compound/DOX dose ratio of 10/1. A positive control was performed by a 10-mM H_2_O_2_ exposure, and a negative control was carried out with cells only receiving PBS. After incubation, a DCFH-DA solution was added in each well to reach a final concentration of 100 µM for 30 min at 37 °C and in darkness. Intensity of the fluorescence was read with a Glomax Explorer (Promega, Leiden, The Netherlands) plate reader at 490/510 nm (excitation/emission). Data were expressed as the relative means of the ROS level.

### 2.7. Cells Exposure and Samples Collection for Metabonomic Study

H9C2 cells were randomly exposed to DOX 0.3 µM, DEX 3 µM, SUC 3 µM and *cis*-ES 3 µM in T-175 flasks for 24 h. Cells were also exposed once to each cardioprotective compound 30 min before DOX exposure, keeping a protective compound/DOX dose ratio of 10/1. After exposure, extracellular media samples were collected and stored at −80 °C. Then, the cell mat was washed twice with PBS and collected in 6 mL of cold methanol by scraping. Samples were then rapidly frozen in liquid nitrogen and stored at −80 °C [19].

### 2.8. Samples Preparation for ^1^H-NMR, Data Acquisition and Treatment

A chloroform-methanol-water extraction procedure was carried out on cells to extract intracellular metabolites [20]. Briefly, a cell lysis by sonication was performed, and the 3 solvents were added to separate the cell contents into an aqueous phase containing hydrophilic metabolites and a chloroform phase containing lipophilic metabolites. The methanol-water phases were collected and evaporated with a speed vacuum concentrator. As previously described, 700 µL of a phosphate-buffered solution (0.2-M Na_2_HPO_4_/0.04-M NaH_2_PO_4_, pH 7.4) were used to dissolve the hydrophilic metabolites. Samples were centrifuged at 10,000× *g* during 10 min. Fifty microliters of trimethylsilylpropanoic acid (TSP) 3.5 mM was added to the supernatant (650 µL) in NMR tubes. Extracellular media samples (500 µL) were mixed with a phosphate-buffered solution (250 µL). Samples were centrifuged at 10,000× *g* during 10 min. Fifty microliters of trimethylsilylpropanoic acid (TSP) 14 mM was added to the supernatant (650 µL) in NMR tubes [17].

^1^H-NMR spectra of both intracellular and extracellular compartments were acquired by a Bruker Advance 600 MHz spectrometer (Bruker BioSpin GmbH, Kontich, Belgium). The employed sequence was NOESYPRESAT-1D, with a number of 256 scans. The acquired free induction decay signals were Fourier-transformed to obtain spectra. Baseline and phase corrections were performed using MestReNova 10.0.2 (Mestrelab Research, Santiago de Compostela, Spain). Spectra were arbitrarily calibrated by setting the TSP-arising resonance to 0.00 ppm. Region from 0.08 to 10 ppm was divided into subareas of 0.04-ppm widths that were further integrated. The water peak (from 4.50 to 5.00 ppm) was suppressed. A total area normalization was carried out for each subarea integral using Excel functionalities (Microsoft Office^®^ 16.36, Redmond, WA, USA).

### 2.9. Multivariate Data Analysis, Metabolites Identification and Statistical Tests

Metabonomic data were analyzed by a projection to a latent structure discriminant analysis (PLS-DA), where the experimental groups were defined as classes. SIMCA P+ 12 (Umetrics, Umeå, Sweden) was used for this purpose. R^2^_cum_ and Q^2^_cum_ parameters, as well as the *p*-value of ANOVA of the cross-validated residuals (CV-ANOVA), were worked out. A cross-validation using 200 permutations was performed to ensure the suitability of the model. The variables characterized by a variable importance in projection (VIP) value > 0.8 were selected as the most discriminant ones, according to recommendations in [21]. Corresponding metabolites were identified with several databases: the Human Metabolome Database (HMDB) [22], Chenomx Profiler software 8.3 (Edmonton, Canada) [23] and “in-house” databases.

Due to the descriptor size of 0.04 ppm that could contain several metabolite chemical shifts but, also, the low signal-to-noise ratio of some metabolites, a semi-quantification comparison of the spectra was then processed to specify the metabolic changes. Statistical significance of the discriminant metabolites was assessed by integrating the ^1^H-NMR peaks of each metabolite with a VIP value > 0.8. Each integral was normalized to the spectral total area, as previously mentioned [17]. The most appropriate statistical tests to compare variables between the different conditions were chosen to respect the two restrictive hypotheses, allowing the use of the parametric tests: the normal distribution of the variable and the equality of the variances. Normality of data was evaluated by a Shapiro-Wilk test [24]. Homoscedasticity of variances was evaluated by a Bartlett’s test [25]. For variables characterized by a normal distribution and a variance homoscedasticity, statistical significance was determined using one-way ANOVA. For other variables, significance was determined using the Dunn’s test. For proportion comparisons, a two-proportion z-test was applied. The significance was determined at *p*-value * < 0.05, ** *p*-value < 0.01 and *p*-value *** < 0.001. Heatmaps for both intracellular and extracellular compartments were established from the mean normalized integrals of the discriminant metabolites, as previously described [17].

To highlight the most relevant metabolic pathways, an enrichment analysis was carried out on the discriminant metabolites. The MetaboAnalyst 4.0. online software was used for this purpose. A metabolite set enrichment analysis (MSEA) was performed [26]. The MSEA is a tool designed to help the interpretation of metabonomic data by highlighting the most relevant biological pathways linked to the signature identified through our analysis. The analysis provides a classification of suggested pathways depending on the number of imputed metabolites found in the identified pathways. A depending *p*-value was worked out and compared to the 0.05 alpha value threshold.

### 2.10. MCF-7 Cell Proliferation Assay

The effects of DOX and protective compounds DEX, SUC and *cis*-ES alone or in combination with DOX on MCF-7 cell proliferation were evaluated by the crystal violet assay. For this purpose, the living cells population was measured at different time points after exposure (days 1, 3, 6, 8 and 9). Exposure medium was renewed every 3 days. Relative mean of the living cells population was determined for each condition.

## 3. Results

### 3.1. GPR91 Expression

GPR91 expression was evaluated on H9C2 and MCF-7 cells by indirect immunofluorescence. GPR91 is diffusedly expressed on H9C2 cells, while no expression was found for MCF-7 (Figure 1). No nonspecific signal was found for the secondary AB staining.

### 3.2. Cell Viability

After 24 h of exposure, no significant change was observed on H9C2 cell viability for all exposure conditions (Figure 2A). After 48 h of exposure, DOX induced a significant decrease of cell viability, and both DEX-DOX and SUC-DOX conditions were not different from the DOX group, while only the *cis*-ES-DOX group described a significant increase of cell viability compared to the DOX group (Figure 2A). After 72 h of exposure, the DEX-DOX, SUC-DOX and *cis*-ES-DOX groups were characterized by a significant increase of cell viability compared to the DOX group. However, those conditions remained different from the control conditions, suggesting that DEX, SUC and *cis*-ES induced a partial beneficial effect on the cell viability (Figure 2A). After 48 h of exposure, all doses (3, 6, 30, 100 µM) of SUC or *cis*-ES were responsible for the same partial benefit on cell viability as compared to DOX alone (Figure 2B,C). Moreover, a combination between DEX with SUC or *cis*-ES did not add any substantial benefit on the cell viability after 48 h of exposure (Figure 2D,E).

### 3.3. Caspase 3 Activity

When H9C2 was exposed to 5 µM of DOX during 4 h, the caspase 3 activity was statistically elevated compared to the control group. The pre-exposure to DEX, SUC and *cis*-ES prior to DOX exposure induced a significant decrease of caspase 3 activity in comparison to the DOX group. However, these conditions remained statistically different from the control group, indicating a partial effect of these compounds against DOX-induced caspase 3 overactivation (Figure 3A). Our findings highlighted a more beneficial yet partial effect for *cis*-ES than DEX against caspase 3 overactivation induced by DOX. Caspase 3 activity was statistically enhanced in the SUC group compared to the control (CTR), although to a lower extent as compared to DOX-induced caspase 3 activation.

### 3.4. Oxidative Stress

When H9C2 was exposed to 5 µM of DOX during 2 h, the ROS level was statistically elevated compared to the control group. Pre-exposing the cells to DEX, SUC and *cis*-ES prior to DOX induced a significant decrease of ROS concentration in comparison to DOX alone. However, these conditions remained statistically different from the control group, indicating a partial effect of these compounds against DOX-induced caspase 3 overactivation (Figure 3B). The results highlighted that *cis*-ES had the most beneficial partial effect against ROS production induced by DOX.

### 3.5. Metabonomic Investigation

^1^H-NMR metabolic signatures of DOX exposure and DEX/SUC/*cis*-ES preincubation on H9C2 cells were obtained after 24 h of exposure, using a PLS-DA multivariate data analysis on both the intra- and extracellular compartment spectra. The scores plot of the intracellular contents indicated a clear discrimination between the metabolic effects caused by DEX and by both receptor agonists (SUC and *cis*-ES), which induced close metabolic responses (Figure 4A). Moreover, the DOX group was closer to the CTR group, while the DEX-DOX/SUC-DOX/*cis-*ES-DOX groups were, respectively, closer to their corresponding conditions without DOX. This latter finding indicated that the DEX and GPR91 agonists induced a bigger metabolic response than DOX itself on H9C2 cells. The PLS-DA model built from intracellular fluid samples was validated by a cross-validation repeated 200 times (Figure 4B). Variables with a VIP score >0.8 were considered as the most discriminant ones, and the corresponding metabolites were identified (Table 1).

The scores plot of the extracellular media samples also indicated a similar discrimination between the metabolic effects caused by DEX and both agonists (SUC and *cis*-ES), suggesting similar metabolic responses from the agonists but distinct from the DEX one (Figure 4C). Moreover, the DOX group was close to the CTR group, as well as the DEX-DOX/SUC-DOX/*cis-*ES-DOX groups were close, respectively, to the DEX/SUC/*cis*-ES groups, indicating that the DEX and GPR91 agonists induced a bigger metabolic response than DOX itself on H9C2 cells. The PLS-DA model on the extracellular media samples was validated by a cross-validation repeated 200 times (Figure 4D). Variables with a VIP score > 0.8 were considered as the most discriminant ones, and the corresponding metabolites were identified (Table 2).

Heatmaps were constructed using normalized areas under the curve (AUC) from the identified discriminant metabolites for both the extra- and intracellular compartments. In each case, the value is the relative mean of the metabolite normalized AUC (Figure 5A,B).

All identified discriminant metabolites were then imputed into the MetaboAnalyst 4.0 online software for a metabolite set enrichment analysis (MSEA). This analysis highlighted the most likely metabolic pathways suggested by the ^1^H-NMR metabonomic data (Figure 6A). Thereby, glucose, alanine, ammonia, glutamate, glutathione, carnitine, sphingolipid metabolisms and the Warburg effect were suggested to be strongly involved in the metabonomic profiles of H9C2 cells exposed to DOX or/and cardioprotective compounds. The enrichment analysis also pointed out several amino acid metabolisms (glycine, serine, arginine, proline, methionine and cysteine); phosphatidylcholine biosynthesis; galactose metabolism; pyruvate metabolism and phospholipid biosynthesis, as well as glycolysis and citric acid cycle. A glutamine-to-glutamate ratio (Gln:Glu) was calculated using the mean normalized AUC for the intracellular content. All groups exposed to SUC or *cis*-ES were characterized by a higher ratio approaching 1, while the other conditions were characterized by a lower ratio under 0.5 (Figure 6B). An extracellular lactate-to-pyruvate ratio (L:P) was worked out using the mean normalized AUC. The DOX condition was characterized by a higher ratio, and all conditions exposed to SUC, *cis*-ES or DEX were characterized by a lower ratio.

### 3.6. MCF-7 Proliferation

The effect of DEX, SUC and *cis*-ES, alone or in combination with DOX, on the proliferation of MCF-7 cells was evaluated using proliferation curves. Only DOX (6 µM) prevented cell proliferation and induced progressive mortality in the population of MCF-7 cells (Figure 7A). The addition of DEX, SUC or *cis*-ES combined with DOX did not affect its ability to induce mortality in MCF-7 cells (Figure 7B). Alone, DEX, SUC and *cis*-ES did not alter the proliferation of MCF-7. However, there was a significant sporadic increase on day 1 in the population of MCF-7 cells in the groups exposed to one of the three cardioprotective agents, compared to the control group. At the following points, the curves of the cardioprotective compounds merged with the curve of the control group, suggesting the absence of an effect of the cardioprotective agents on the normal proliferation of MCF-7.

## 4. Discussion

H9C2 cells are derived from a rat cardiomyoblast cell line similar to differentiated cardiomyocytes but without any cardiac functional characteristics. This cellular model was employed due to the constraints imposed by ^1^H-NMR spectroscopy that require more than 10 million cells per sample to ensure a correct signal-to-noise ratio. For this purpose, the ability of H9C2 cells to reproduce the main features of the DOX toxic mode of action, such as oxidative stress [27,28], apoptosis [29,30], topoisomerases inhibition and sarcoplasmic reticulum stress [31,32], were ensured through a literature review and our own experimental data. Despite some obvious drawbacks, the H9C2 cell line remains a suitable model for studying DOX-induced cardiotoxicity and cardioprotective strategies. However, a further investigation of the GPR91 agonist-induced protective effect should be carried out on more human extrapolated in vitro/in vivo models for validation.

In a previous work, we noticed an oversecretion of SUC in the extracellular medium when H9C2 cells were co-exposed to DEX and DOX during 24 h. We hypothesized that DEX could possibly protect against DOX-induced cardiotoxicity through an auto/paracrine stimulation of the GPR91 receptor by SUC, triggering the cell survival pathways [17]. In this study, the protective effects of SUC and *cis-*ES, two GPR91 agonists, against DOX-induced cardiotoxicity were investigated on H9C2 cells. For long periods of exposure (24 to 72 h), DOX caused a decrease of cell viability, an effect partially counteracted by both SUC and *cis*-ES pre-exposures. Interestingly, the protection obtained with both agonists was close to the one already seen for DEX pre-exposure, the reference cardioprotective compound. Moreover, on a short time of exposure (4 h), DOX significantly triggered the apoptotic process by an enhanced activity of the caspase 3 enzyme, as already reported [33]. SUC pre-exposure was able to decrease the activity of caspase 3 to a similar extent as DEX pre-exposure, whereas *cis*-ES pre-exposure provoked an even more pronounced effect. The important elevation of ROS levels caused by DOX was partially prevented by SUC and DEX to a similar extent, whereas *cis*-ES showed a more drastic effect. These results suggest that agonism of the GPR91 receptor protects H9C2 cells against DOX-induced toxicity, especially when using *cis*-ES as the agonist. *cis*-ES was used in this study because of its higher specificity and affinity to GPR91 than SUC [34] and to avoid any possible additional effects of SUC that are also involved in cell respiration.

In our conditions, a pre-exposure to GPR91 agonists inhibited apoptosis induced by DOX. There are some evidences linking GPR91 to both apoptosis and cell growth. Indeed, some authors reported that GPR91 induces cardiac hypertrophy [15,35], while others observed that a prolonged incubation of cardiomyocytes with high concentrations of succinate (10 mM) promoted apoptosis [16]. Similar observations were made in the context of an ischemic injury, where the release of succinate and the triggering of the GPR91-associated signaling pathway were responsible for mitochondrial dysfunction and apoptosis in cardiomyocytes [36]. In the present study, although both agonists significantly reduced the DOX-induced overactivity of caspase 3, SUC alone induced an increased activity of caspase 3, while no such effect was found for *cis*-ES alone. Those findings suggest that the proapoptotic or the pro-growth effect mediated by GPR91 could depend on the exposure context, and exposure conditions and should be further investigated.

It has been reported that succinate dehydrogenase (SDH), forming complex II of the respiratory chain in mitochondria, has an important role in redox regulation in the heart as an enhancer or suppressor of ROS. Some authors claim that SUC is an important regulator of SDH-induced ROS production and that a low level of SUC is associated with a higher ROS production by SDH [37,38,39]. The SUC level seems to be a critical factor for the ROS regulation performed by SDH. In our study, we observed a decreased ROS level associated with a higher level of SUC, as detected by ^1^H-NMR, when H9C2 cells were pre-exposed to SUC or *cis*-ES before adding DOX, as compared to DOX alone. Those results suggest that the decrease of the ROS level may be caused by the higher level of SUC promoting the ROS suppression activity of SDH. The link between GPR91 activation by agonists and the regulation of the ROS level by SDH is still unclear. Additional investigations are needed to demonstrate if SUC and *cis-*ES reduce the ROS level trough a GPR91-dependent or -independent manner.

The expression of GPR91 was validated in H9C2 cells by immunofluorescence staining, in accordance to similar observations by other authors [36], suggesting that this receptor may be strongly implicated in the protective effects of *cis*-ES and SUC. However, to ensure and validate the crucial role of the GPR91 receptor in the observed protective effects, a receptor knockout should be performed, as already reported [40,41].

In a previous study, we characterized the metabolic changes induced by DOX and DEX in H9C2 cells [17]. In this study, we focused on the metabolic effects of a 30-min preincubation with SUC or *cis*-ES before 24 h-DOX exposure on H9C2 cells using a ^1^H-NMR-based metabonomic approach. Both the intra- and extracellular compartments of H9C2 cells were investigated for this purpose. The metabonomic profiles obtained under such conditions pointed out that the main metabolic differences were due to cardioprotective compounds rather than DOX exposure. SUC and *cis*-ES exposures and pre-exposure had similar metabolic profiles, which were different from the DEX-induced profiles. The enrichment analysis using MetaboAnalyst software highlighted many pathways that are related to energy metabolism and anabolism. Identified metabolite variations strongly suggest that both SUC and *cis*-ES stimulate aerobic metabolism mainly through glycolysis and the Krebs cycle. The extracellular level of glucose is decreased in all SUC and *cis*-ES conditions, suggesting a higher uptake of glucose from the culture medium. A strong increase in extracellular pyruvate and a small increase in intra- and extracellular lactate concentrations are also observed in all SUC and *cis*-ES conditions, but the extracellular L:P remains higher than those of the CTR and DOX conditions. The L:P ratio is commonly used in serums to detect mitochondrial disorders, as it reflects the equilibrium between the product and substrate of the reaction catalyzed by lactate dehydrogenase and indirectly reflects the NADH:NAD^+^ redox status of the intracellular compartment [42,43]. A high L:P ratio is associated with dysfunction of the mitochondrial respiratory chain, where there is an increase in reducing equivalents (excess of NADH and absence of NAD^+^) [44]. Therefore, this ratio can be used as an indicator of the cell aerobic or anaerobic status. Here, we transposed the L:P ratio to the culture fluid that mimics the blood compartment. As previously observed, DOX induced a metabolic switch to anaerobic glycolysis [17], characterized by a high L:P ratio. All protective compounds induced a decrease of the L:P ratio compared to the DOX and CTR conditions. This strongly suggests that SUC, *cis*-ES and DEX stimulate the mitochondrial respiratory chain, resulting in a stronger aerobic metabolism. Besides glycolysis, the Krebs cycle can also use other substrates for adenosine triphosphate (ATP) production. The metabonomic results highlight that glycolysis was used as the main pathway to supply the Krebs cycle when the cells were exposed to the GPR91 agonists. Indeed, the intracellular content of carnitine was lowered in such conditions, suggesting that fatty acid oxidation is poorly used for ATP production. Moreover, the intracellular detection of glutamine and glutamate reflects that glutamine present in the culture medium entered into cells where it was converted into glutamate by a glutaminase enzyme. Glutamate can be used as fuel for the Krebs cycle. The intracellular Gln:Glu ratio was calculated to evaluate the activity of glutaminase. This ratio was higher in all SUC and *cis*-ES conditions, assuming that glutamine is poorly converted into glutamate, as a consequence of a lower need for glutamate to supply ATP production. The Gln:Glu ratio was already used to evaluate the dependence of cancer cells to glutamine for energy purposes and as an indicator of tissue oxygenation [45,46]. All SUC and *cis*-ES conditions were characterized by a lower intracellular level of proline, a precursor of α-ketoglutarate, to supply the citric cycle. Thus, a lower level of proline may highlight that this amino acid is less-produced, highlighting a lower need to form α-ketoglutarate. The results also indicate a higher level of isoleucine in the extracellular compartment of cells exposed to GPR91 agonists, as compared to the CTR. Isoleucine is an essential amino acid that cannot be produced by cells and must be supplied by the culture medium. A higher extracellular concentration reflects a lower consumption by cells. Despite the fact that isoleucine can be used as a “non-glucose” substrate for the Krebs cycle, its lower consumption is also an argument that SUC and *cis*-ES stimulate the aerobic energy metabolism using glucose as the main fuel. An increase in the SUC level was observed in the extracellular compartment of cells exposed to GPR91 agonists. When SUC was used as the agonist (at a nondetectable concentration by ^1^H-NMR), this increase was bigger than the one observed with DEX exposure. An enhancement of the SUC level strongly indicates a stimulation of the Krebs cycle in the mitochondria for energy production that is induced by GPR91 agonists. Moreover, the observations suggest that SUC is secreted in the extracellular compartment and could stimulate the GPR91 receptor to sustain the primary effect of SUC and *cis*-ES on this receptor. This hypothesis could explain why an exposure with a very low concentration of SUC and *cis*-ES (3 µM) is able to ensure a cardioprotective effect that is not enhanced with a higher exposure concentration as a “plateau effect”. Some evidences suggest that SUC may impair the pyruvate dehydrogenase (PDH) activity through the GPR91-dependent and -independent signaling pathways in cardiomyocytes suffering from an ischemia/reperfusion injury [47]. Hence, the association between high extracellular levels of SUC and pyruvate might be linked to this phenomenon. However, as previously discussed, the low L:P ratio observed in cells exposed to either GPR91 agonists or DEX strongly suggests an aerobic mitochondrial metabolism for energy production and a lower production of lactate from pyruvate. Therefore, the effect of the GPR91 agonists on the PDH activity should be further investigated to elucidate this phenomenon in the context of DOX-induced cardiotoxicity.

The metabonomic data also highlighted an important SUC and *cis*-ES-induced stimulation of phospholipids synthesis, although to different extents, as seen with DEX. First, both the phosphocholine (PCho) and glycerophosphocholine (GPC) levels were enhanced in cells exposed to GPR91 agonists but to a lower extent than DEX, indicating an ongoing activation of the Kennedy’s pathway responsible for phosphatidylcholine synthesis, a major phospholipid found in biological membranes [48]. Moreover, the levels of choline and its precursor glycine were decreased in such conditions compared to the DEX pre-exposure effect, also supporting that the activation of the Kennedy’s pathway by SUC and *cis*-ES remains less intense in DEX conditions. A similar observation was made about phosphatidylserine synthesis. Indeed, the serine level, the precursor of this phospholipid, was elevated in SUC and *cis*-ES conditions, again to a lower extent than in DEX conditions. However, SUC seems to more actively stimulate phosphatidylinositol synthesis than DEX. Indeed, the intracellular level of *myo*-inositol, the precursor of phosphatidylinositol, was higher with the agonists than with DEX. Therefore, our results strongly suggest that both GPR91 agonists were able to stimulate phospholipids biosynthesis to counteract DOX-induced cardiotoxicity, although to a lower extent than DEX does.

Glutathione is a tripeptide composed of glutamate, glycine and cysteine. It is found in all mammal tissues and is a powerful agent for redox homeostasis maintenance, especially in conditions where cells could be damaged by oxidative stress [49]. DOX induced a harmful oxidative stress in H9C2 that can be partially prevented by pre-exposing cells to either SUC or *cis*-ES. The metabonomic results pointed out a decrease of glutathione and precursors (glycine and glutamate) induced by the GPR91 agonists. Moreover, our results showed a decrease of the intracellular taurine and carnitine contents. Considering that those compounds can be used as antioxidants to counteract DOX-induced oxidative stress [50,51], their decreased concentration could indicate a lower cell need for redox maintenance. It sounds as if both GPR91 agonists induced a decrease of oxidative stress using other mechanisms than glutathione, taurine and carnitine synthesis. As already discussed, the increase of the SUC level is suspected to reduce the ROS production induced by SDH in mitochondria and should be responsible for a lower need of antioxidant compounds such as glutathione, carnitine and taurine for redox maintenance. Oxidative stress was considered for long as the main DOX-induced toxicological mechanism causing mitochondrial dysfunction and cell death. Oxidative stress was therefore seen as the “starting point” of all cell damages induced by DOX in the heart. However, both SUC and *cis*-ES GPR91 agonists were able to reduce DOX-induced oxidative stress, while those compounds have no known antioxidant properties. This observation, together with the lack of protective effect reported for many antioxidants but DEX, contributes to questioning the role of oxidative stress in DOX-induced cardiotoxicity. Indeed, among all the tested antioxidants, only DEX was able to exhibit a cardioprotective effect in the clinical trial. Thereafter, some authors reported that DEX was also able to prevent the DOX-induced inhibition of the mitochondrial topoisomerase IIβ [9]. Considering these evidences, it appears that DOX-induced oxidative stress should be considered as a “consequence” rather than a “cause” in the cardiotoxicological process. Our metabonomic data strongly suggest that the synthesis of endogenous antioxidants is decreased in cells pre-exposed to GPR91 agonists. Both agonists may act on the cause(s) of the oxidative stress rather than on the oxidative stress itself, which could then be targeted for the development of cardioprotective strategies.

So far, there have been no clinical trial evaluating the benefit of a GPR91 agonist or antagonist against any particular pathology, as well as there being no drug candidate involving SUC or *cis*-ES as an active ingredient. All discovered GPR91 agonists and antagonists are still experimental molecules only available for research purposes. Considering that more than 50% currently approved drugs target G-protein coupled receptors, there are many hopes that drug candidates targeting GPR91 will be developed. In our in vitro study, we highlighted that targeting GPR91 with an agonist could partially counteract DOX-induced cardiotoxicity. However, further investigations are needed for studying GPR91 agonists’ efficacy and safety in the relevant animal models before considering clinical studies. As GPR91 agonists are expected to activate GPR91 in all cells expressing it, some optimization would be necessary to ensure the heart-selective delivery and to avoid adverse effects due to interactions on other GPR91-expressing organs. In this context, tissue-selective drug delivery systems such as liposomes or specific vectors may be developed and evaluated in rigorous preclinical studies.

As breast cancer is one of the main indications for DOX treatment, human breast cancer-derived MCF-7 cells were selected to assess the effect of a cardioprotective strategy targeting the GPR91 receptor on cancer cell growth and on DOX pharmacological action. Our results demonstrate that such agonists do not interfere with the DOX anticancer effects on MCF-7 cells, alone or in combination. GPR91 expression was also undetected in MCF-7 cells. Based on those findings, one can reasonably assume that targeting the GPR91 receptor for cardioprotective purposes should not interfere with the DOX pharmacological effects on cancer cells that do not express the GPR91 receptor. Although the possible involvement of GPR91 and SUC in tumor development is not clearly established, some works suggest that SUC and its receptor may be tumor promoters. It has been reported that the intracellular concentration of SUC, as well as other metabolites such as fumarate, aspartate and 2-hydroxyglutarate, is increased in some cancers and is responsible for the epigenetic changes associated with carcinogenesis and impacting the activity of proteins involved in cell signaling and metabolism [52,53]. The accumulation of SUC in cancer cells is secondary to mutations in SDH causing a loss-of-function of the enzyme, which no longer converts SUC to fumarate. These mutations are sufficient to promote the development of some cancers, including renal carcinoma [54]. There is currently no evidence of a direct link between the GPR91 receptor and carcinogenesis. Nevertheless, the known implications of the receptor in processes found during tumor development such as angiogenesis [55], fibrosis [56] and inflammation [57] raise questions about its possible participation in the development of some cancers. In addition, a recent study pointed out that some cancer cells, whose SDH activity was reduced, were able to secrete SUC in the tumor microenvironment, causing the activation of macrophages to “tumor-associated macrophages” (TAM) that play a role in tumor progression and metastases developments [58]. The presence of TAM in the tumor microenvironment is indeed associated with a poor prognosis in breast cancer, ovarian cancer and in some types of gliomas and lymphomas [59]. For the development of a cardioprotective therapy targeting the GPR91 receptor to counteract the adverse effects of anticancer drugs, special attention should be paid to a possible adverse effect promoting the development of cancers. Further studies should determine whether the expression of the GPR91 receptor in cancer cells, the secretion of succinate by the tumor and the presence of TAM in the tumor microenvironment could be exclusive criteria or not for a cardioprotective strategy targeting this receptor.

## 5. Conclusions

In this study, SUC and *cis*-ES, two agonists of the GPR91 receptor, were investigated as cardioprotective compounds to counteract the DOX-induced treatment-limiting cardiotoxicity in H9C2 cells expressing GPR91. A pre-exposure with each agonist was able to decrease partially the mortality, the oxidative stress and the apoptosis induced by DOX. Moreover, a ^1^H-NMR-based metabonomics assessment highlighted several metabolic changes associated with the cardioprotective effects of both agonists: aerobic metabolism stimulation using glucose as the main fuel (low L:P ratio and high Gln:Glu ratio), phospholipids synthesis, redox maintenance and succinate excretion. Finally, the exposure of breast cancer MCF-7 cells to each agonist with/without DOX had no effect on the MCF-7 proliferation and DOX pharmacological effect. Therefore, targeting the GPR91 receptor is an interesting strategy to counteract DOX-induced cardiotoxicity and should be further investigated.

## Figures and Tables

**Figure 1 cells-09-02177-f001:**
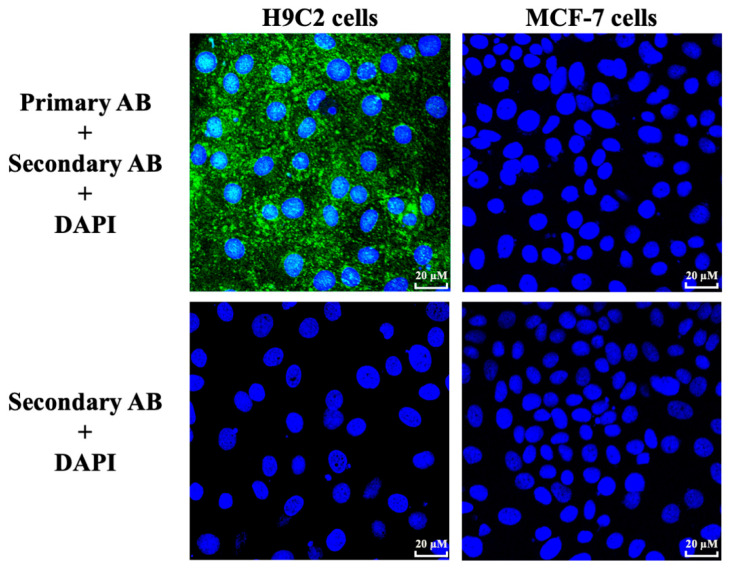
Indirect immunofluorescence staining of the GPR91 receptor on H9C2 and MCF-7 cells. Cells were labeled with 4′,6-diamidino-2-phénylindole (DAPI) and rabbit anti-GPR91 antibody (AB) revelated with fluorescent Alexa Fluor 488 secondary AB. A control without primary AB was made for both cell lines.

**Figure 2 cells-09-02177-f002:**
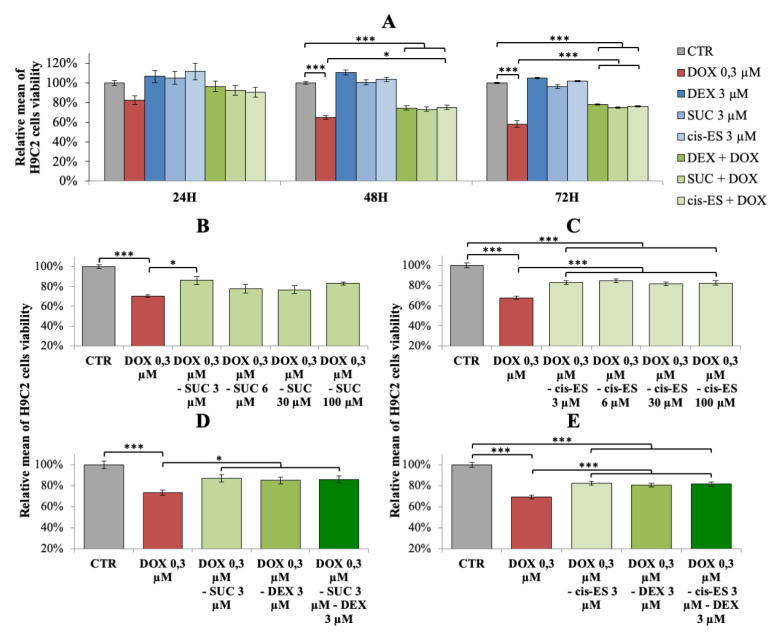
Relative mean (+/−SEM) of the H9C2 cell viability. (**A**) Exposure (24, 48 and 72 h) to doxorubicin (DOX) and cardioprotective compounds administrated alone or 30 min prior to DOX exposure. (**B**) Exposure (48 h) to DOX alone and with increasing doses of succinate (SUC). (**C**) Exposure (48 h) to DOX alone and with increasing doses of *cis-*epoxysuccinate (*cis*-ES)**.** (**D**) Exposure (48 h) to DOX alone and with a combination of dexrazoxane (DEX) and SUC. (**E**) Exposure (48 h) to DOX alone and with a combination of DEX and *cis*-ES. ANOVA one-way with multiple comparisons: * *p*-value < 0.05 and *** *p*-value < 0.001. CTR: control.

**Figure 3 cells-09-02177-f003:**
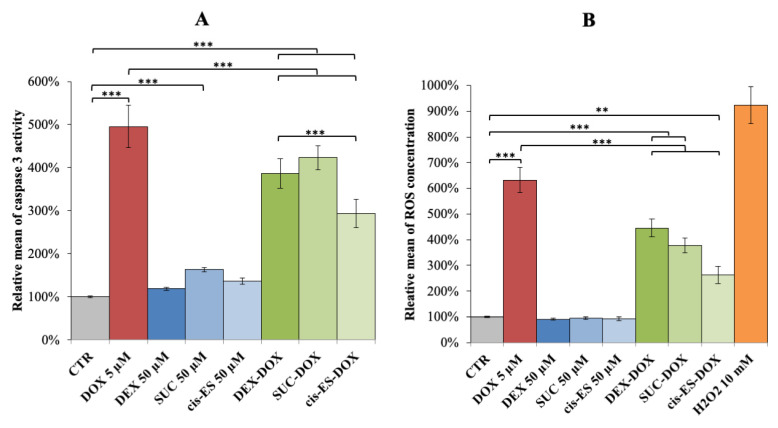
(**A**) Relative mean (+/−SEM) of caspase 3 activity after 4 h of exposure to DOX and cardioprotective compounds administrated alone or 30 min prior to DOX exposure. (**B**) Relative mean (+/−SEM) of the reactive oxygen species (ROS) concentration after 2 h of exposure to DOX and cardioprotective compounds administrated alone or 30 min prior to DOX exposure. ANOVA one-way with multiple comparisons: ** *p*-value < 0.01 and *** *p*-value < 0.001.

**Figure 4 cells-09-02177-f004:**
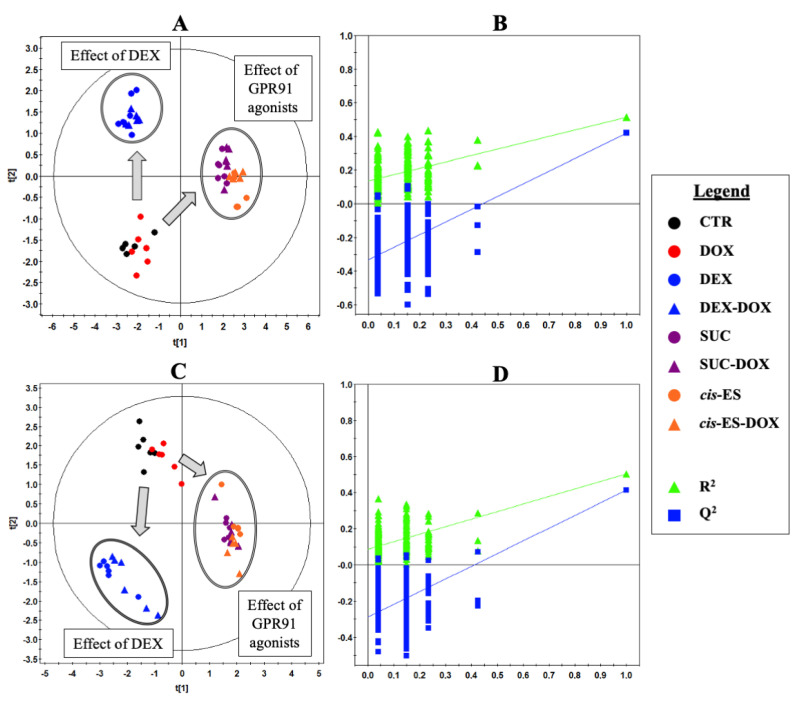
Projection to a latent structure discriminant analysis (PLS-DA) of the metabonomic data. (**A**) Scores plot of intracellular extracts after 24 h of exposure. R^2^_cum_ = 0.58, Q^2^_cum_ = 0.48, Hotelling’s T2 = 0.95 and *p*-value of ANOVA of the cross-validated residuals (CV-ANOVA) < 0.001. (**B**) Cross-validation plot of the intracellular extracts (R^2^ in green and Q^2^ in blue) with a permutation test repeated 200 times. The *Y*-axis intercepts were R^2^ = (0.0; 0.12) and Q^2^ = (0.0; −0.32). (**C**) Scores plot of the extracellular compartments after 24 h of exposure. R^2^_cum_ = 0.50, Q^2^_cum_ = 0.41, Hotelling’s T2 = 0.95 and *p*-value (CV-ANOVA) < 0.001. (**D**) Cross-validation plot of the extracellular compartments (R^2^ in green and Q^2^ in blue) with a permutation test repeated 200 times. The *Y*-axis intercepts were R^2^ = (0.0; 0.09) and Q^2^ = (0.0; −0.29).

**Figure 5 cells-09-02177-f005:**
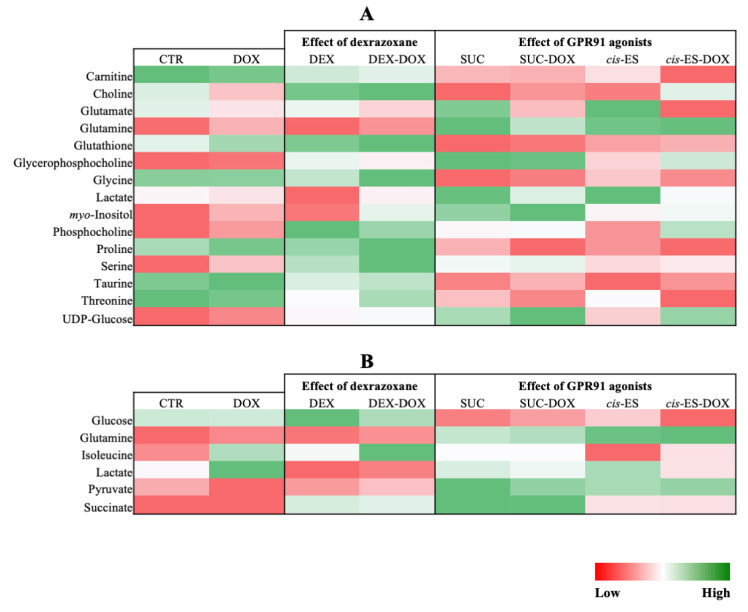
Heatmaps constructed from the relative means of the discriminant metabolite normalized integrals. (**A**) Intracellular compartment. (**B**) Extracellular compartment.

**Figure 6 cells-09-02177-f006:**
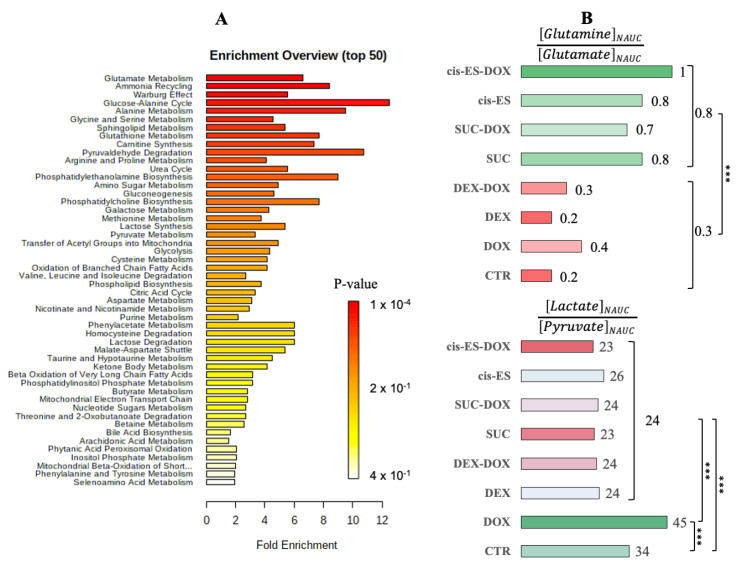
(**A**) Metabolite set enrichment analysis performed on the ^1^H-NMR metabonomic data, using the MetaboAnalyst 4.0 online software. (**B**) ^1^H-NMR intracellular glutamine-to-glutamate and extracellular lactate-to-pyruvate ratios calculated using the normalized integrals. Two-proportion z-test: *** *p*-value < 0.001.

**Figure 7 cells-09-02177-f007:**
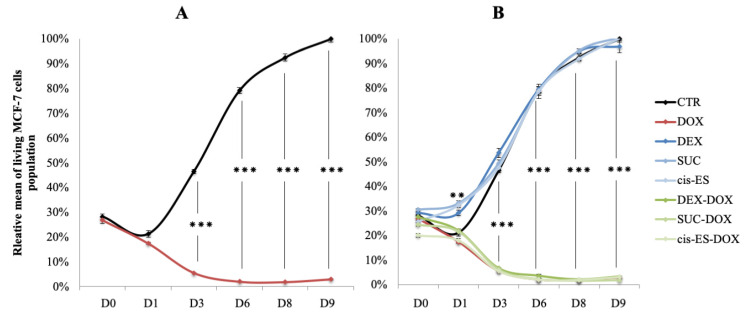
Relative mean (+/−SEM) of living MCF-7 cells population after day 0, 1, 3, 6, 8 and 9 of the first exposure to DOX, and cardioprotective compounds administrated alone or 30 min prior to DOX exposure. (**A**) Proliferation curves from the CTR and DOX-exposed cells. (**B**) Proliferation curves from the CTR, DOX-exposed and protective compound-exposed cells. Student’s *t*-test: ** *p*-value < 0.01 and *** *p*-value < 0.001.

**Table 1 cells-09-02177-t001:** Identified discriminant metabolites for the intracellular compartments, with the corresponding chemical shifts.

Metabolites (VIP)	Chemical Shifts (ppm)	Effect of DEX on DOX-Exposed Cells	Effect of SUC and *cis*-ES on DOX-Exposed Cells
Carnitine (2.82)	33.22 (s) 3.40 (s)	↓	↓ ***
Choline (2.82)	3.20 (s)	↑ *	↓
Glutamate (3.15)	2.10 (m) 2.36 (m) 3.77 (m)	↓	↓
Glutamine (5.71)	2.12 (m) 2.45 (m) 3.76 (m)	↓	↑ **
Glutathione (4.96)	2.15 (m) 2.55 (m) 3.00 (t)	↑	↓ *
Glycerophosphocholine (2.82)	3.24 (s)	↑	↑ *
Glycine (5.66)	3.57 (s)	↑	↓ **
Lactate (0.97)	1.32 (d) 4.12 (q)	↑	↑
*myo*-Inositol (1.44)	3.52 (m) 3.60 (m) 4.07 (t)	↑	↑ *
Phosphocholine (2.82)	3.23 (s)	↑ ***	↑ *
Proline (1.54)	2.01 (m) 2.35 (m) 3.33 (m) 4.10 (m)	↑	↓ **
Serine (3.28)	3.95 (m) 4.00 (m)	↑ ***	↑
Taurine (6.09)	3.25 (t) 3.43 (t)	↓	↓ ***
Threonine (0.97)	1.32 (d) 4.25 (m)	↓	↓ **
Uridine diphosphate glucose (1.58)	3.50 (m) 3.80 (m) 4.20 (m) 5.98 (t) 7.95 (d)	↑	↑ ***

Peaks multiplicity is indicated between the parentheses (s = singulet, d = doublet, t = triplet and m = multiplet). Arrows indicate metabolite concentration changes compared to pre-exposure samples: ↑ means an increased level, and ↓ means a decreased level. Variable importance in the projection (VIP) values are specified between parentheses. Dunn’s test: * *p*-value < 0.05, ** *p*-value < 0.01 and *** *p*-value < 0.001. DEX: dexrazoxane, DOX: doxorubicin, SUC: succinate and *cis*-ES: *cis-*epoxysuccinate.

**Table 2 cells-09-02177-t002:** Identified discriminant metabolites for the extracellular compartments, with the corresponding chemical shifts.

Metabolites (VIP)	Chemical Shifts (ppm)	Effect of DEX on DOX-Exposed Cells	Effect of SUC and *cis*-ES on DOX-Exposed Cells
Glucose (4.02)	3.23 (dd) 3.39 (m) 3.45 (m) 3.52 (dd) 3.72 (m)	↑	↓ *
	3.82 (m) 3.88 (dd) 5.22 (d)		
Glutamine (5.19)	2.12 (m) 2.44 (m) 3.76 (t)	=	↑ ***
Isoleucine (2.23)	0.94 (t) 1.04 (d) 1.26 (m) 1.45 (m) 3.65 (d)	↑	↓
Lactate (6.44)	1.32 (d) 4.12 (q)	↓ ***	↓ *
Pyruvate (1.40)	2.38 (s)	↑	↑ ***
Succinate (2.23)	2.40 (s)	↑ **	↑ **

Peaks multiplicity is indicated between parentheses (s = singulet, d = doublet, t = triplet and m = multiplet). Arrows indicate metabolite concentration changes compared to the pre-exposure samples: ↑ means an increased level, ↓ means a decreased level and = means an identical level. Variable importance in the projection (VIP) values are specified between parentheses. Dunn’s test: * *p*-value < 0.05, ** *p*-value < 0.01 and *** *p*-value < 0.001.

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
