# Peer review of "GPR91 Receptor Mediates Protection against Doxorubicin-Induced Cardiotoxicity without Altering Its Anticancer Efficacy. An In Vitro Study on H9C2 Cardiomyoblasts and Breast Cancer-Derived MCF-7 Cells"

_cells, 2020, doi:10.3390/cells9102177_

Round 1
Reviewer 1 Report
In this manuscript, the authors reported the results of an experimental study evaluating the impact of succinate and cis-epoxysuccinate in the prevention of doxorubicin cardiotoxicity.
In general, the authors should be congratulated for their work and for their choice of this important topic. The main limitation for the use of anthracyclines is cardiovascular effect and risk of cardiac damage. And as cancer is a burden, increasing the knowledge in anthracycline-induced cardiotoxicity might help specialists to improve the management of the disease and make it possible to use chemotherapy drugs safely.
The objective of the present work was to study the effect of succinate (SUC) and cis-epoxysuccinate (cis-ES), two agonists of GPR91 25 receptor, on DOX-induced cardiotoxicity to H9C2 cells. Both SUC and cis-ES decreased partially the cellular mortality, 33 the oxidative stress and the apoptosis induced by DOX. It is an mechanistic study with potential relevance.
However, the following issues should be addressed to evaluate the relevance of the manuscript. .
- Is there any study proposed to evaluate these substances in the clinical setting? The authors should include discussion or data supporting the possible clinical use of these drugs in phase I studies.
- If you decide to do a study succinate and cis-epoxysuccinate in cardiotoxicity prevention in a clinical trial, how would the trial be conducted in phase I or II?
- Is there a drug that a patient can take orally that contains these substances used in the study?
- The authors should discuss the next steps following this study.
Author Response
We want to acknowledge the reviewer for his congratulations and his pertinent remarks. As required, we added a paragraph in the discussion section about the clinical prospects.
Until now there have been no clinical trial using GPR91 agonists (SUC, cis-ES or other compound) and currently there is no available drugs containing these compounds as active ingredients. GPR91 agonists or antagonists are still experimental compounds for research purpose only. Preclinical assessments using relevant animal models are needed for studying GPR91 agonists safety, efficacy and their possible optimization (ie avoiding adverse effects on other tissues expressing GPR91, ensuring an optimal heart distribution) before performing clinical trials. In the current state of knowledge, a clinical trial is not yet possible. We hope that our work will generate interest in order to achieve this step.
Reviewer 2 Report
Re: GPR91 receptor mediates protection against doxorubicin induced-cardiotoxicity without altering its anticancer efficacy. An in vitro study on H9C2 cardiomyoblasts and breast cancer-derived MCF-7 cells
This study was based on previous data published recently, showing that GPR91 receptor plays a role in dexrazoxane cardioprotective mechanisms. Here the authors examined the protective effects of two agonists of GPR91 receptor, succinate (SUC) and cis-epoxysuccinate (cis-ES), against doxorubicin-induced cardiotoxicity and compared their effects with dexrazoxane. The results showed that both SUC and cis-ES reduced the oxidative stress and cell death of cardiomyocytes, similar to the effects of dexrazoxane. Further metabonomics showed that both GPR91 agonists stimulated aerobic metabolism with glucose as main fuel and phospholipid synthesis, which. The GPR91 agonists did not affect proliferation of MCF-7 breast cancer cells
The study was well planned and well presented. The conclusions are supported by the results. Overall, this is an interesting work, which provides baseline for further studies.
Author Response
We want to acknowledge the reviewer for his general opinion.